# Deep Predictive Coding Network with Local Recurrent Processing for Object Recognition

**Kuan Han[1,3], Haiguang Wen[1,3], Yizhen Zhang[1,3], Di Fu[1,3], Eugenio Culurciello[1,2], Zhongming Liu[1,2,3]\***

[1]School of Electrical and Computer Engineering, Purdue University
[2]Weldon School of Biomedical Engineering, Purdue University
[3]Purdue Institute for Integrative Neuroscience, Purdue University

## Abstract

Inspired by "predictive coding" - a theory in neuroscience, we develop a bi-directional and dynamic neural network with local recurrent processing, namely predictive coding network (PCN). Unlike feedforward-only convolutional neural networks, PCN includes both feedback connections, which carry top-down predictions, and feedforward connections, which carry bottom-up errors of prediction. Feedback and feedforward connections enable adjacent layers to interact locally and recurrently to refine representations towards minimization of layer-wise prediction errors. When unfolded over time, the recurrent processing gives rise to an increasingly deeper hierarchy of non-linear transformation, allowing a shallow network to dynamically extend itself into an arbitrarily deep network. We train and test PCN for image classification with SVHN, CIFAR and ImageNet datasets. Despite notably fewer layers and parameters, PCN achieves competitive performance compared to classical and state-of-the-art models. Further analysis shows that the internal representations in PCN converge over time and yield increasingly better accuracy in object recognition. Errors of top-down prediction also reveal visual saliency or bottom-up attention.

## 1 Introduction

Modern computer vision is mostly based on feedforward convolutional neural networks (CNNs) [18, 33, 50]. To achieve better performance, CNN models tend to use an increasing number of layers [19, 24, 50, 59], while sometimes adding "short-cuts" to bypass layers [19, 56]. What motivates such design choices is the notion that models should learn a deep hierarchy of representations to perform complex tasks in vision [50, 59]. This notion generally agrees with the brain's hierarchical organization [31, 62, 67, 16]: visual areas are connected in series to enable a cascade of neural processing [60]. If one layer in a model is analogous to one area in the visual cortex, the state-of-the-art CNNs are considerably deeper (with 50 to 1000 layers) [20, 18] than the visual cortex (with 10 to 20 areas) . As we look to the brain for more inspiration, it is noteworthy that biological neural networks support robust and efficient intelligence for a wide range of tasks without any need to grow their depth or width [37].

What distinguishes the brain from CNNs is the presence of abundant feedback connections that link a feedforward series of brain areas in reverse order [11]. Given both feedforward and feedback connections, information passes not only bottom-up but also top-down, and interacts with one another to update the internal states over time. The interplay between feedforward and feedback connections

has been thought to subserve the so-called "predictive coding" [44, 14, 52, 25, 12] - a neuroscience theory that becomes popular. It says that feedback connections from a higher layer carry the prediction of its lower-layer representation, while feedforward connections in turn carry the error of prediction upward to the higher layer. Repeating such bi-directional interactions across layers renders the visual system a dynamic and recurrent neural network [1, 12]. Such a notion can also apply to artificial neural networks. As recurrent processing unfolds in time, a static network architecture is used over and over to apply increasingly more non-linear operations to the input, as if the input were computed through more and more layers stacked onto an increasingly deeper feedforward network [37]. In other words, running computation through a bi-directional network for a longer time may give rise to an effectively deeper network to approximate a complex and nonlinear transformation from pixels to concepts [37, 6], which is potentially how brain solves invariant object recognition without the need to grow its depth.

Inspired by the theory of predictive coding, we propose a bi-directional and dynamical network, namely *Deep Predictive Coding Network* (PCN), to run a cascade of local recurrent processing [30, 5, 43] for object recognition. PCN combines predictive coding and local recurrent processing into an iterative inference algorithm. When tested for image classification with benchmark datasets (CIFAR-10, CIFAR-100, SVHN and ImageNet), PCN uses notably fewer layers and parameters to achieve competitive performance relative to classical or state-of-the-art models. Further behavioral analysis of PCN sheds light on its computational mechanism and potential use for mapping visual saliency or bottom-up attention.

## 2   Related Work

**Predictive Coding** In the brain, connections between cortical areas are mostly reciprocal [11]. Rao and Ballard suggest that bi-directional connections subserve "predictive coding" [44]: feedback connections from a higher cortical area carry neural predictions to the lower cortical area, while the feedforward connections carry the unpredictable information (or error of prediction) to the higher area to correct the neuronal states throughout the hierarchy. With supporting evidence from empirical studies [1, 52, 28], this mechanism enables iterative inference for perception[44] and unsupervised learning[53], incorporates modern neural networks for classification [54] and video prediction [39], and likely represents a unified theory of the brain[12, 25].

**Predictive Coding Network with Global Recurrent Processing** Driven by the predictive coding theory, a bi-directional and recurrent neural network has been proposed in [63]. It runs global recurrent processing by alternating a bottom-up cascade of feedforward computation and a top-down cascade of feedback computation. For each cycle of recurrent dynamics, the feedback prediction starts from the top layer and propagates layer by layer until the bottom layer; then, the feedforward error starts from the bottom layer and propagates layer by layer until the top layer. The model described herein is similar, but uses local recurrent processing, instead of global recurrent processing. Only for the convenience of notation in this paper, we refer to the proposed PCN with local recurrent processing simply as "PCN", while referring to the model in [63] explicitly as "PCN with global recurrent processing".

**Local Recurrent Processing** In the brain, feedforward-only processing plays a central role in rapid object recognition[47, 9]. Although less understood, feedback connections are thought to convey top-down attention [4, 3] or prediction [12, 44, 52]. Evidence also suggests that feedback signals may operate between hierarchically adjacent areas along the ventral stream [30, 5, 43] to enable local recurrent processing for object recognition [65, 2], especially given ambiguous or degraded visual input [64, 51]. Therefore, feedback processes may be an integral part of both global and local recurrent processing underlying top-down attention in a slower time scale and visual recognition in a faster time scale.

## 3   Predictive Coding Network

Herein, we design a bi-directional (feedforward and feedback) neural network that runs local recurrent processing between neighboring layers, and we refer to this network as Predictive Coding Network (PCN). As illustrated in Fig. 1, PCN is a stack of recurrent blocks, each running dynamic and recurrent processing within itself through feedforward and feedback connections. Feedback connections are

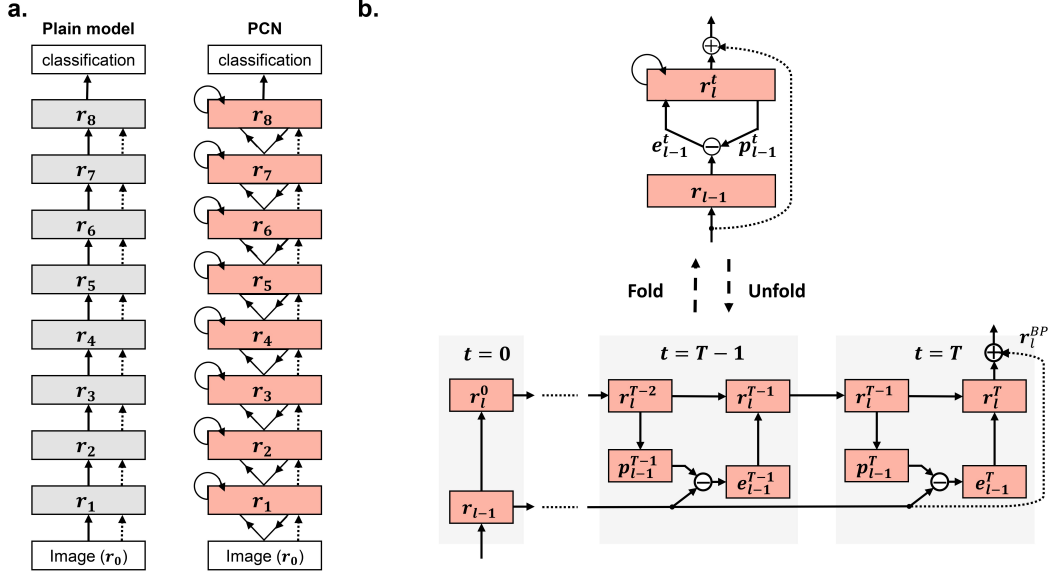

Figure 1: Architecture of CNN vs. PCN (a) The plain model (left) is a feedforward CNN with 3×3 convolutional connections (solid arrows) and 1×1 bypass connections (dashed arrows). On the basis of the plain model, the local PCN (right) uses additional feedback (solid arrows) and recurrent (circular arrows) connections. The feedforward, feedback and bypass connections are constructed as convolutions, while the recurrent connections are constructed as identity mappings (b) The PCN consists of a stack of basic building blocks. Each block runs multiple cycles of local recurrent processing between adjacent layers, and merges its input to its output through the bypass connections. The output from one block is then sent to its next block to initiate local recurrent processing in a higher block. It continues until reaching the top of the network.

used to predict lower-layer representations. In turn, feedforward connections send the error of prediction to update the higher-layer representations. After repeating this processing for multiple cycles within a given block, the lower-layer representation is merged to the higher-layer representation through a bypass connection. The merged representation is further sent as the input to the next recurrent block to start another series of recurrent processing in a higher level. After the local recurrent processing continues through all recurrent blocks in series, the emerging top-level representations are used for image classification.

In the following mathematical descriptions, we use italic letters as symbols for scalars, bold lowercase letters for column vectors and bold uppercase letters for matrices. We use $T$ to denote the number of recurrent cycles, $\boldsymbol{r}_l(t)$ to denote the representation of layer $l$ at time $t$, $\boldsymbol{W}_{l-1,l}$ to denote the feedforward weights from layer $l-1$ to layer $l$, $\boldsymbol{W}_{l,l-1}$ to denote the feedback weights from layer $l$ to layer $l-1$ and $\boldsymbol{W}_{l-1,l}^{bp}$ to denote the weights of bypass connections.

## 3.1 Local Recurrent Processing in PCN

Within each recurrent block (e.g. between layer $l-1$ and layer $l$), the local recurrent processing serves to reduce the error of prediction. As in Eq. (1), the higher-layer representation $\boldsymbol{r}_l(t)$ generates a prediction, $\boldsymbol{p}_{l-1}(t)$, of the lower-layer representation, $\boldsymbol{r}_{l-1}$, through feedback connections, $\boldsymbol{W}_{l,l-1}$, yielding an error of prediction $\boldsymbol{e}_{l-1}(t)$ as Eq. (2).

$$\boldsymbol{p}_{l-1}(t) = \left(\boldsymbol{W}_{l,l-1}\right)^T \boldsymbol{r}_l(t) \tag{1}$$
$$\boldsymbol{e}_{l-1}(t) = \boldsymbol{r}_{l-1} - \boldsymbol{p}_{l-1}(t) \tag{2}$$

The objective of recurrent processing is to reduce the sum of the squared prediction error (Eq. (3)) by updating the higher-layer representation, $\boldsymbol{r}_l(t)$, with an gradient descent algorithm [55]. In each cycle of recurrent processing, $\boldsymbol{r}_l(t)$ is updated along the direction opposite to the gradient (Eq. (4)) with an

incremental size proportional to an update rate, $\alpha_l$. As $\boldsymbol{r}_l(t)$ is updated over time as Eq. (5), it tends to converge while the gross error of prediction tends to decrease. Note that Eq. (6) is equivalent to Eq. (5), if the feedback weights are tied to be the transpose of the feedforward weights. Eq. (6) is useful even without this assumption as shown in a prior study [63], and it is thus used in this study instead of Eq. (5).

$$L_{l-1}(t) = \frac{1}{2} \parallel \boldsymbol{r}_{l-1} - \boldsymbol{p}_{l-1}(t) \parallel_2^2 \tag{3}$$

$$\frac{\partial L_{l-1}(t)}{\partial \boldsymbol{r}_l(t)} = -\boldsymbol{W}_{l,l-1}\boldsymbol{e}_{l-1}(t) \tag{4}$$

$$\boldsymbol{r}_l(t+1) = \boldsymbol{r}_l(t) - \alpha_l \frac{\partial L_{l-1}(t)}{\partial \boldsymbol{r}_l(t)} = \boldsymbol{r}_l(t) + \alpha_l \boldsymbol{W}_{l,l-1}\boldsymbol{e}_{l-1}(t) \tag{5}$$

$$\boldsymbol{r}_l(t+1) = \boldsymbol{r}_l(t) + \alpha_l (\boldsymbol{W}_{l-1,l})^T \boldsymbol{e}_{l-1}(t) \tag{6}$$

### 3.2 Network Architecture

We implement PCN with some architectural features common to modern CNNs. Specifically, feedforward and feedback connections are implemented as regular convolutions and transposed convolutions [10], respectively, with a kernel size of 3. Bypass convolutions are implemented as $1 \times 1$ convolution. Batch normalization (BN) [26] is applied to the input to each recurrent block. During the recurrent processing between layer $l-1$ and layer $l$, rectified linear units (ReLU) [41] are applied to the initial output $\boldsymbol{r}_l(0)$ at $t = 0$ and the prediction error at each time step $\boldsymbol{e}_{l-1}(t)$, as expressed by Eq. (7) and Eq. (8), respectively. ReLU renders the local recurrent processing an increasingly non-linear operation as the processing continues over time.

$$\boldsymbol{r}_l(0) = ReLU\left((\boldsymbol{W}_{l-1,l}^T \boldsymbol{r}_{l-1})\right) \tag{7}$$

$$\boldsymbol{e}_{l-1}(t) = ReLU\left(\boldsymbol{r}_{l-1} - \boldsymbol{p}_{l-1}(t)\right) \tag{8}$$

Besides, a $2 \times 2$ max-pooling with a stride of 2 is optionally applied to the output from every 2 (or 3) blocks. On top of the highest recurrent block is a classifier, including a global average pooling, a fully-connected layer, followed by softmax.

For comparison, we also design the feedforward-only counterpart of PCN and refer to it as the "plain" model. It includes the same feedforward and bypass connections and uses the same classification layer as in PCN, but excludes any feedback connection or recurrent processing. The architecture of the plain model is similar to that of Inception CNN [59].

---

**Algorithm 1** Predictive Coding Network with local recurrent processing.

---

**Input:** The input image $\boldsymbol{r}_0$;
 1: **for** $l = 1$ to $L$ **do**
 2:     $\boldsymbol{r}_{l-1}^{BN} = BatchNorm(\boldsymbol{r}_{l-1})$;
 3:     $\boldsymbol{r}_l(0) = ReLU\left(FFConv\left(\boldsymbol{r}_{l-1}^{BN}\right)\right)$;
 4:     **for** $t = 1$ to $T$ **do**
 5:         $\boldsymbol{p}_{l-1}(t) = FBConv\left(\boldsymbol{r}_l(t-1)\right)$;
 6:         $\boldsymbol{e}_{l-1}(t) = ReLU\left(\boldsymbol{r}_{l-1} - \boldsymbol{p}_{l-1}(t)\right)$;
 7:         $\boldsymbol{r}_l(t) = \boldsymbol{r}_l(t-1) + \alpha_l FFconv\left(\boldsymbol{e}_{l-1}(t)\right)$;
 8:     **end for**
 9:     $\boldsymbol{r}_l = \boldsymbol{r}_l(T) + BPConv\left(\boldsymbol{r}_{l-1}^{BN}\right)$;
10: **end for**
11: **return** $\boldsymbol{r}_L$ for classification;
12: ▷ FFConv represents the feedforward convolution, FBConv represents the feedback convolution and BPConv represents the bypass convolution.

---

Our implementation is described in Algorithm 1. Note that the update rate $\alpha_l$ used for local recurrent processing is a learnable and non-negative parameter separately defined for each filter in each recurrent block. The number of cycles of recurrent processing - an important parameter in PCN, is varied to be $T = 1, ..., 5$. For both PCN and its CNN counterpart (i.e. T=0), we design multiple

architectures (labeled as A through E) suitable for different benchmark datasets (SVHN, CIFAR and ImageNet), as summarized in Table 1. For example, PCN-A-5 stands for a PCN with architecture A and 5 cycles of local recurrent processing.

Table 1: Architectures of PCN. Each column shows a model. We use PcConv to represent a predictive coding layer, with its parameters denoted as "PcConv<kernel size>-<number of channels in feedback convolutions>-<number of channels in feedforward convolutions>". The first layer in PCN-E is a regular convolution with a kernel size of 7, a padding of 3, a stride of 2 and 64 output channels. "*" indicates the layer applying maxpooling to its output. Feature maps in one grid have the same size.

| PCN Configuration | | | | | |
|---|---|---|---|---|---|
| Dataset | SVHN | | CIFAR | | ImageNet |
| Architecture | A | B | C | D | E |
| #Layers | 7 | | 9 | | 13 |
| Image Size | $32 \times 32$ | | | | $224 \times 224$ |
| Layers | PcConv3-3-16 PcConv3-16-16 | PcConv3-3-16 PcConv3-16-32 | PcConv3-3-64 PcConv3-64-64 | PcConv3-3-64 PcConv3-64-64 | Conv7-64 PcConv3-64-64 |
| | PcConv3-16-32* PcConv3-32-32 | PcConv3-32-64* PcConv3-64-64 | PcConv3-64-128* PcConv3-128-128 | PcConv3-64-128* PcConv3-128-128 | PcConv3-64-128* PcConv3-128-128 |
| | PcConv3-32-64* PcConv3-64-64 | PcConv3-64-128* PcConv3-128-128 | PcConv3-128-256* PcConv3-256-256 | PcConv3-128-256* PcConv3-256-256 | PcConv3-128-128* PcConv3-128-128 |
| | | | PcConv3-256-256 PcConv3-256-256 | PcConv3-256-512 PcConv3-512-512 | PcConv3-128-256* PcConv3-256-256 PcConv3-256-256 |
| | | | | | PcConv3-256-512* PcConv3-512-512 PcConv3-512-512 |
| Calssification | global average pooling, FC-10/100/1000, softmax | | | | |
| #Params | 0.15M | 0.61M | 4.91M | 9.90M | 17.26M |

## 4  Experiments

We train PCN with local recurrent processing and its corresponding plain model for object recognition with the following datasets, and compare their performance with classical or state-of-the-art models.

### 4.1  Datasets

**CIFAR** The CIFAR-10 and CIFAR-100 datasets [32] consist of $32 \times 32$ color images drawn from 10 and 100 categories, respectively. Both datasets contain 50,000 training images and 10,000 testing images. For preprocessing, all images are normalized by channel means and standard derivations. For data augmentation, we use a standard scheme (flip/translation) as suggested by previous works [19, 23, 38, 45, 24].

**SVHN** The Street View House Numbers (SVHN) dataset [42] consists of $32 \times 32$ color images. There are 73,257 images in the training set, 26,032 images in the test set and 531,131 images for extra training. Following the common practice [23, 38], we train the model with the training and extra sets and tested the model with the test set. No data augmentation is introduced and we use the same pre-processing scheme as in CIFAR.

**ImageNet** The ILSVRC-2012 dataset [7] consists of 1.28 million training images and 50k validation images, drawn from 1000 categories. Following [19, 20, 59], we use the standard data augmentation scheme for the training set: a $224 \times 224$ crop is randomly sampled from either the original image or its horizontal flip. For testing, we apply a single crop or ten crops with size $224 \times 224$ on the validation set and the top-1 or top-5 classification error is reported.

### 4.2  Training

Both PCN and (plain) CNN models are trained with stochastic gradient descent (SGD). On CIFAR and SVHN datasets, we use 128 batch-size and 0.01 initial learning rate for 300 and 40 epochs, respectively. The learning rate is divided by 10 at 50%, 75% and 87.5% of the total number of training epochs. Besides, we use a Nesterov momentum of 0.9 and a weight decay of 1e-3, which is determined by a 45k/5k split on the CIFAR-100 training set. On ImageNet, we follow the most common practice [19, 66] and use a initial learning rate of 0.01, a momentum of 0.9, a weight decay

Table 2: Error rates (%) on CIFAR datasets. #L and #P are the number of layers and parameters, respectively.

| Type | Model | #L | #P | C-10 | C-100 |
|---|---|---|---|---|---|
| Feed-Forward Models | HighwayNet [56] | 19 | 2.3M | 7.72 | 32.39 |
| | FractalNet [34] | 21 | 38.6M | 5.22 | 23.30 |
| | ResNet [19] | 110 | 1.7M | 6.41 | 27.22 |
| | ResNet [20] | 164 | 1.7M | 5.23 | 24.58 |
| | (Pre-act) | 1001 | 10.2M | 4.62 | 22.71 |
| | WRN [69] | 28 | 36.5M | 4.00 | 19.25 |
| | DenseNet | 100 | 0.8M | 4.51 | 22.27 |
| | BC [24] | 190 | 25.6M | **3.46** | **17.18** |
| Recurrent Models | RCNN [36] | 160 | 1.86M | 7.09 | 31.75 |
| | DasNet [58] | - | - | 9.22 | 33.78 |
| | FeedbackNet [70] | 12 | | - | 28.88 |
| | CliqueNet | 18 | 10.14M | 5.06 | 23.14 |
| | [68] | 30 | 10.02M | **5.06** | **21.83** |
| | PCN with | 7 | 0.57M | 7.60 | 31.69 |
| | Global Recurrent | 9 | 1.16M | 7.20 | 30.66 |
| | Processing [63] | 9 | 4.65M | 6.17 | 27.42 |
| PCN | PCN-C-1 | 9 | 4.91M | 5.70 | 24.01 |
| | PCN-C-2 | 9 | 4.91M | 5.38 | 22.89 |
| | PCN-C-5 | 9 | 4.91M | 5.10 | 22.43 |
| | PCN-D-1 | 9 | 9.90M | 5.73 | 23.78 |
| | PCN-D-2 | 9 | 9.90M | 5.39 | 22.75 |
| | PCN-D-5 | 9 | 9.90M | **4.89** | **21.77** |
| Plain Models | Plain-C | 9 | 2.59M | 5.68 | 25.65 |
| | Plain-D | 9 | 5.21M | 5.61 | 25.31 |

Table 3: Error rates (%) on SVHN. #L and #P are the number of layers and parameters, respectively.

| Model | #L | #P | SVHN |
|---|---|---|---|
| MaxOut [15] | - | - | 2.47 |
| NIN [38] | - | - | 2.35 |
| DropConnect [61] | - | - | 1.94 |
| DSN [35] | - | - | 1.92 |
| RCNN [36] | 6 | 2.67M | 1.77 |
| FitNet [45] | 13 | 1.5M | 2.42 |
| WRN [69] | 16 | 11M | 1.54 |
| PCN (Global) | 7 | 0.14M | 2.42 |
| [63] | 7 | 0.57M | 2.42 |
| PCN-A-1 | 7 | 0.15M | 2.29 |
| PCN-A-2 | 7 | 0.15M | 2.22 |
| PCN-A-5 | 7 | 0.15M | 2.07 |
| PCN-B-1 | 7 | 0.61M | 1.99 |
| PCN-B-2 | 7 | 0.61M | 1.97 |
| PCN-B-5 | 7 | 0.61M | 1.96 |
| Plain-A | 7 | 0.08M | 2.85 |
| Plain-B | 7 | 0.32M | 2.43 |

of 1e-4, 100 epochs with the learning rate dropped by 0.1 at epochs 30, 60 & 90. The batch size is 256 for PCN-E-0/1/2, 128 for PCN-E-3/4, 115 for PCN-E-5 due to limited computational resource.

### 4.3 Evaluating the Behavior of PCN

To understand how the model works, we further examine how local recurrent processing changes the internal representations of PCN. For this purpose, we focus on testing PCN-D-5 with CIFAR-100.

**Converging representation?** Since local recurrent processing is governed by predictive coding, it is anticipated that the error of prediction tends to decrease over time. To confirm this expectation, the L2 norm of the prediction error is calculated for each layer and each cycle of recurrent processing and averaged across all testing examples in CIFAR-100. This analysis reveals the temporal behavior of recurrently refined internal representations.

**What does the prediction error mean?** Since the error of prediction drives the recurrent update of internal representations, we exam the spatial distribution of the error signal (after the final cycle) first for each layer and then average the error distributions across all layers by rescaling them to the same size. The resulting error distribution is used as a spatial pattern in the input space, and it is applied to the input image as a weighted mask to visualize its selectivity.

**Does predictive coding help image classification?** The goal of predictive coding is to reduce the error of top-down prediction, seemingly independent of the objective of categorization that the PCN model is trained for. As recurrent processing progressively updates layer-wise representation, does each update also subserve the purpose of categorization? As in [27], the loss of categorization, as a nonlinear function of layer-wise representation $\mathcal{L}(\boldsymbol{r}_l(t))$, shows its Taylor's expansion as Eq. (9), where $\Delta\boldsymbol{r}_l(t) = \boldsymbol{r}_l(t+1) - \boldsymbol{r}_l(t)$ is the incremental update of $\boldsymbol{r}_l$ at time $t$.

$$\mathcal{L}(\boldsymbol{r}_l(t+1)) - \mathcal{L}(\boldsymbol{r}_l(t)) = \Delta\boldsymbol{r}_l(t) \cdot \frac{\partial \mathcal{L}(\boldsymbol{r}_l(t))}{\partial \boldsymbol{r}_l(t)} + \mathcal{O}(\Delta\boldsymbol{r}_l(t)^2) \qquad (9)$$

If each update tends to reduce the categorization loss, it should satisfy $\Delta\boldsymbol{r}_l(t) \cdot \frac{\partial \mathcal{L}(\boldsymbol{r}_l(t))}{\partial \boldsymbol{r}_l(t)} < 0$, or the "cosine distance" between $\Delta\boldsymbol{r}_l(t)$ and $\frac{\partial \mathcal{L}(\boldsymbol{r}_l(t))}{\partial \boldsymbol{r}_l(t)}$ should be negative. Here $\mathcal{O}(\cdot)$ is ignored as suggested by [27], since the incremental part becomes minor because of the convergent representation in PCN. We test this by calculating the cosine distance for each cycle of recurrent processing in each layer, and then averaging the results across all testing images in CIFAR-100.

Table 4: Error rates (%) on ImageNet.

| Model | #Layers | #Params | Single-Crop | | 10-crop | |
|---|---|---|---|---|---|---|
| | | | top-1 | top-5 | top-1 | top-5 |
| ResNet-18 | 18 | 11.69M | 30.24 | 10.92 | 28.15 | 9.40 |
| ResNet-34 | 34 | 21.80M | 26.69 | 8.58 | 24.73 | 7.46 |
| ResNet-50 | 50 | 25.56M | 23.87 | 7.14 | 22.57 | 6.24 |
| PCN-E-5 | 13 | 17.26M | 25.31 | 7.79 | 23.52 | 6.64 |
| PCN-E-3 | | | 25.36 | 7.78 | 23.62 | 6.69 |
| Plain-E | 13 | 9.34M | 31.15 | 11.27 | 28.82 | 9.79 |

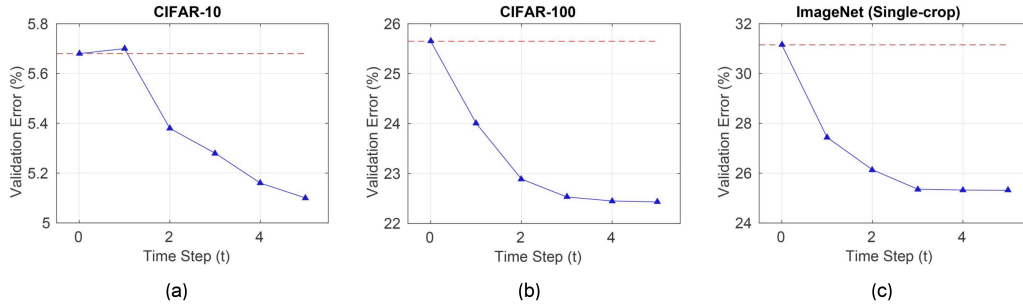

Figure 2: PCN shows better categorization performance given more cycles of recurrent processing, for CIFAR-10, CIFAR-100 and ImageNet. The red dash line represents the accuracy of the plain model.

## 4.4 Experimental results

**Classification performance** On CIFAR and SVHN datasets, PCN always outperforms its corresponding CNN counterpart (Table 2 and 3). On CIFAR-100, PCN-D-5 reduces the error rate by 3.54% relative to Plain-D. The performance of PCN is also better than those of the ResNet family [19, 20], although PCN is much shallower with only 9 layers whereas ResNet may use as many as 1001 layers. Although PCN under-performs WRN [69] and DenseNet-190 [24] by 2-4%, it uses a much shallower architecture with many fewer parameters. On SVHN, PCN shows competitive performance despite fewer layers and parameters. On ImageNet, PCN also performs better than its plain counterpart. With 5 cycles of local recurrent processing (i.e. PCN-E-5), PCN slightly under-performs ResNet-50 but outperforms ResNet-34, while using fewer layers and parameters than both of them. Therefore, the classification performance of PCN compares favorably with other state-of-the-art models especially in terms of the performance-to-layer ratio.

**Recurrent Cycles** The classification performance generally improves as the number of cycles of local recurrent processing increases for both CIFAR and ImageNet datasets, and especially for ImageNet (Fig. 2). It is worth noting that this gain in performance is achieved without increasing the number of layers or parameters, but by simply running computation for longer time on the same network.

**Local vs. Global Recurrent Processing** The PCN with local recurrent processing (proposed in this paper) performs better than the PCN with global recurrent processing (proposed in our earlier paper [63]. Table 2 shows that PCN with local recurrent processing reduces the error rate by 5% on CIFAR-100 compared to PCN with the same architecture but global recurrent processing. In addition, local recurrent processing also requires less computational resource for both training and inference.

## 4.5 Behavioral Analysis

To understand how/why PCN works, we further examine how local recurrent processing changes its internal representations and whether the change helps categorization. Behavioral analysis reveals some intriguing findings.

As expected for predictive coding, local recurrent processing progressively reduces the error of top-down prediction for all layers, except the top layer on which image classification is based (Fig.

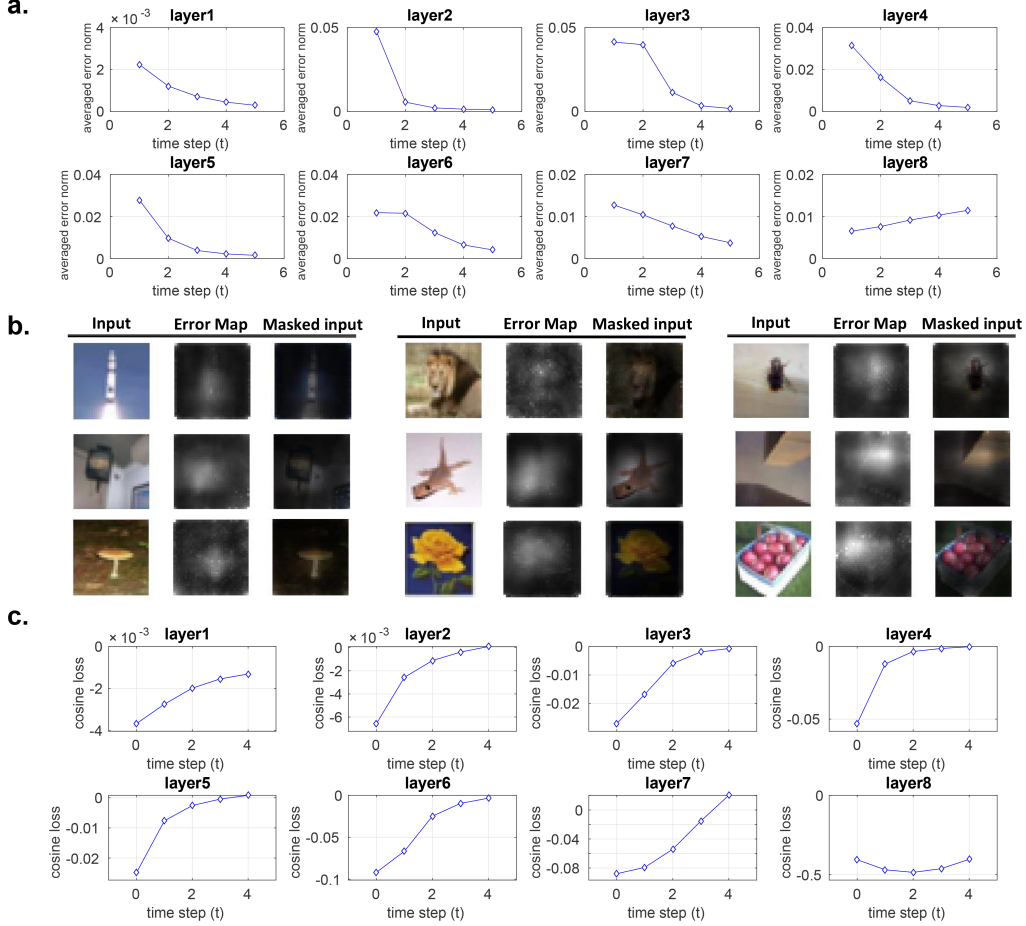

Figure 3: Behavioral analysis of PCN. (a) Errors of prediction tend to converge over repeated cycles of recurrent processing. The norm of the prediction error is shown for each layer and each cycle of local recurrent processing. (b) Errors of prediction reveal visual saliency. Given an input image (left), the spatial distribution of the error averaged across layers (middle) shows visual saliency or bottom-up attention, highlighting the part of the input with defining features (right). (3) The averaged cosine loss between $\Delta r_l(t)$ and $\frac{\partial \mathcal{L}(r_l(t))}{\partial r_l(t)}$.

3a). This finding implies that the internal representations converge to a stable state by local recurrent processing. Interestingly, the spatial distribution of the prediction error (averaged across all layers) highlights the apparently most salient part of the input image, and/or the most discriminative visual information for object recognition (Fig. 3b). The recurrent update of layer-wise representation tends to align along the negative gradient of the categorization loss with respective to the corresponding representation (Fig. 3c). From a different perspective, this finding lends support to an intriguing implication that predictive coding facilitates object recognition, which is somewhat surprising because predictive coding, as a computational principle, herein only explicitly reduces the error of top-down prediction without having any explicit role that favors inference or learning towards object categorization.

## 5    Discussion and Conclusion

In this study, we advocate further synergy between neuroscience and artificial intelligence. Complementary to engineering innovation in computing and optimization, the brain must possess additional mechanisms to enable generalizable, continuous, and efficient learning and inference. Here, we highlight the fact that the brain runs recurrent processing with lateral and feedback connections under

predictive coding, instead of feedforward-only processing. While our focus is on object recognition, the PCN architecture can potentially be generalized to other computer vision tasks (e.g. object detection, semantic segmentation and image caption) [13, 22, 40], or subserve new computational models that can encode [62, 16, 29, 49] or decode [46, 17, 48, 57] brain activities. PCN with local recurrent processing outperforms its counterpart with global recurrent processing, which is not surprising because global feedback pathways might be necessary for top-down attention [4, 3], but may not be necessary for core object recognition [9] itself. By modeling different mechanisms, we support the notion that local recurrent processing is necessary for the initial feedforward process for object recognition.

This study leads us to rethink about the models for classification beyond feedforward-only networks. One interesting idea is to evaluate the equivalence between ResNets and recurrent neural networks (RNN). Deep residual networks with shared weights can be strictly reformulated as a shallow RNN [37]. Regular ResNets can be reformulated as time-variant RNNs [37], and their representations are iteratively refined along the stacked residual blocks [27]. Similarly, DenseNet has been shown as a generalized form of higher order recurrent neural network (HORNN) [6]. The results in this study are in line with such notions: a dynamic and bi-directional network can refine its representations across time, leading to convergent representations to support object recognition.

On the other hand, PCN is not contradictory to existing feedforward models, because the PCN block itself is integrated with a Inception-type CNN module. We expect that other network modules are applicable to further improve PCN performance, including cutout regularization [8], dropout layer [21], residual learning [19] and dense connectivities [24]. Although not explicitly trained, error signals of PCN can be used to predict saliency in images, suggesting that other computer vision tasks [13, 22, 40] could benefit from the diverse feature representations (e.g. error, prediction and state signals) in PCN.

The PCN with local recurrent processing described herein has the following advantages over feedforward CNNs or other dynamic or recurrent models, including a similar PCN with global recurrent processing [63]. 1) It can achieve competitive performance in image classification with a shallow network and fewer parameters. 2) Its internal representations converge as recurrent processing proceeds over time, suggesting a self-organized mechanism towards stability [12]. 3) It reveals visual saliency or bottom-up attention while performing object recognition. However, its disadvantages are 1) the longer time of computation than plain networks (with the same number of layers) and 2) the sequentially executed recurrent processing, both of which should be improved or addressed in future studies.

## Acknowledgement

The research was supported by NIH R01MH104402 and the College of Engineering at Purdue University.

## Footnotes

*Correspondence to: Zhongming Liu <zmliu@purdue.edu>

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
