[Reviews · NeurIPS 2018]

Reviewer 1



This paper presents the predictive coding network (PCN), a convolutional architecture with local recurrent and feedback connections. Higher layers provide top-down predictions while the lower layers provide the prediction errors, which are refined over time by the local recurrence. This idea is not new, other work (such as that of Lotter et al. and others) have used this for other tasks, such as video prediction and object recognition, though this has yet to be shown to scale to larger scale tasks such as ImageNet. The authors compare the performance of PCN, with varying number of cycles of recurrent processing, to standard CNN architectures on multiple image datasets. In general, PCN has slightly lower error than standard architectures with a comparable number of parameters. However, while PCN also generally outperforms their plain model, the plain model always has significantly fewer parameters. It would be beneficial for to include a more in-depth analysis on the effects of the local recurrence and feedback, individually and combined, while controlling for the number of parameters. For instance, in Table 4, the base model has 9.34 million parameters and the PCN has 17.26 million parameters. The correct parameter controlled comparison would actually be between two PCNs but unrolled for fewer timesteps, to get at the question of whether the multi-interaction recurrence is meaningfully deepening the network in time. In fact, the ImageNet top1 performance difference between PCN-E-5 and PCN-E-3 (which share the same number of parameters) does not appear to be significant. Instead, the right comparison to be made explicit in Table 4 should be the distinction in validation error between timesteps 1 and 2 (or all other timesteps > 1) in the rightmost plot (for ImageNet) – the same comparisons should be made in Tables 2 and 3 as well. In general, clarity could also be improved, especially in the model section. Overall, the authors demonstrate improvements on standard image classification datasets using a predictive coding framework.

Reviewer 2



ANN have been extremely simplified with respect to biological networks. This paper makes a good step toward testing more ideas from biology. It is a strong paper as is. I would love to see the authors extend the bio-inspired to the connectivity between layers. Not a set 3x3 convolution rather a wider and stochastic set of connections whose weights are then learned. Thinking about this further it is not clear to me how your skip connection works. As one recurrent pair ping pong back and forth up to four times what is happening to the by pass? Does it wait for the recurrent layers to advance one step before it advances one step? Regardless of the bypass details your network can be viewed as an ensemble of networks side by side where all receive the primary input and the recurrent processed data step from left to right across the networks. A question I have is why no funneling down of the bypass information is an exact copy of the input the best?

Reviewer 3



Summary: This paper proposes a method for object recognition based on the predictive coding principle from neuroscience. They use a recurrent neural network (RNN) in each layer computation of a convolutional neural network (CNN). The RNNs performs bottom-up and top-down inference of the features at each layer to minimize layer-wise prediction errors. In experiments, they authors show competitive performance against state-of-the-art recognition methods, and also outperform a baseline based on global predictive coding rather than the local predictive coding approach used here. Additionally, experiments showing analysis of the recurrent cycles within each layer in relation to classification performance. Pros: Novel local predictive coding based architecture/formulation that improves previous predictive coding based methods Competitive performance in relation to the state-of-the-art recognition models with less parameters. Interesting analysis of the recurrent cycles within the network Comments: Fewer number of layers argument: Throughout the paper the number of layers is used as argument to highlight the advantage of the proposed method. However, doesn’t the fact that there are RNNs within each layer implicitly add up to the number of layer within the total computation graph? When RNNs are being unrolled through time, we are technically unrolling multi-layer network with inputs at each layer and special gates to move into the next layer of computation. Can the authors give their insight on this? Similar number of parameters comparison with baselines: In the comparison tables with the state-of-the-art, it would be nice to have comparisons with the same number of parameters. Either reduce the baselines number of parameters or (if possible) increase the number of parameters of the proposed method. Clarity: In figure 1b, it would be nice if the authors can make it clear that the arrows between the light-gray boxes means the feature is being sent to the next computation block (if I am not mistaken) (e.g., r_l^{T-1} -> r_l^{T-1}). Overall, I like this paper, however, it would be nice if the authors can address the issues brought up by the reviewer. The use of predictive coding for layers to converge to optimal features to improve the recognition task is attractive.